METHODS AND RESOURCES

# A new branched proximity hybridization assay for the quantification of nanoscale protein–protein proximity

Shuangshuang Zheng[1], Melanie Sieder [1,2], Michael Mitterer[3], Michael Reth [1,3], Marco Cavallari[1,3], Jianying Yang [1,3]*

1 BIOSS Centre For Biological Signaling Studies and Department of Molecular Immunology, Biology III, Faculty of Biology, University of Freiburg, Freiburg, Germany, 2 Spemann Graduate School for Biology and Medicine (SGBM), Freiburg, Germany, 3 Max Planck Institute of Immunobiology and Epigenetics, Freiburg, Germany

* jianying.yang@biologie.uni-freiburg.de

## Abstract

Membrane proteins are organized in nanoscale compartments. Their reorganization plays a crucial role in receptor activation and cell signaling. To monitor the organization and reorganization of membrane proteins, we developed a new branched proximity hybridization assay (bPHA) allowing better quantification of the nanoscale protein–protein proximity. In this assay, oligo-coupled binding probes, such as aptamer, nanobody, and antibodies, are used to translate the proximity of target proteins to the proximity of oligos. The closely positioned oligos then serve as a template for a maximum of 400-fold branched DNA (bDNA) signal amplification. The amplified bPHA signal is recorded by flow cytometer, thus enabling proximity studies with high throughput, multiplexing, and single-cell resolution. To demonstrate the potential of the bPHA method, we measured the reorganization of the immunoglobulin M (IgM)- and immunoglobulin D (IgD)-class B cell antigen receptor (BCR) on the plasma membrane and the recruitment of spleen tyrosine kinase (Syk) to the BCR upon B lymphocyte activation.

## Introduction

Previously, it was thought that most receptors are freely diffusing monomers that are activated by ligand binding and cross-linking [1,2]. Recent studies suggest, however, that membrane proteins are confined in nanoscale compartments that are reorganized upon cellular activation [3,4]. This reorganization involves nanometer changes of the proximity of membrane receptors, and proficient methods measuring receptor proximity would be essential to understand better these cellular processes. However, methods to quantify nanoscale protein–protein proximity in an efficient manner are still in their infancy.

Because of the 250-nm diffraction limit of the light, the light microscope cannot monitor nanoscale membrane processes, and this limitation has resulted in the development of several different superresolution microscopy methods. However, these techniques require careful

**Funding:** Funding for this work was provided by Deutsche Forschungsgemeinschaft TRR130-P02 (B-Zellen: Immunität und Autoimmunität, http://www.trr130.forschung.uni-erlangen.de/index.php/en/home.html, MR), Max Planck Society (https://www.ie-freiburg.mpg.de/de, MR and JY), Excellence Initiative of the German Federal and State Governments EXC294 (https://www.bioss.uni-freiburg.de, MR and JY), and Spemann Graduate School for Biology and Medicine (SGBM, https://www.sgbm.uni-freiburg.de, MS). The funders had no role in study design, data collection and analysis, decision to publish, or preparation of the manuscript.

**Competing interests:** I have read the journal's policy and the authors of this manuscript have the following competing interests: patent applications have been filed (US and European Patent Office, World Intellectual Property Organization) by the Campus Technologies Freiburg and the Max-Planck Innovation covering this technology, for which JY, MC, and MR are inventors.

**Abbreviations:** APC, allophycocyanin; APEX, ascorbic acid peroxidase; BCR, B cell antigen receptor; bDNA, branched DNA; BiFC, bimolecular fluorescence complementation; bPHA, branched proximity hybridization assay; BRET, bioluminescence resonance energy transfer; Cas9, CRISPR-associated protein 9; CRISPR, clustered regularly interspaced short palindromic repeats; Cy5, cyanine 5; Enh, Enhancer; Fab, fragment antigen binding; FITC, fluorescein isothiocyanate; FRET, Förster resonance energy transfer; GFP, green fluorescent protein; IgD, immunoglobulin D; IgM, immunoglobulin M; ITAM, immunoreceptor tyrosine-based activation motif; LC, light chain; MFI, mean fluorescence intensity; mIg, membrane-bound immunoglobulin; PE, phycoerythrin; PLA, proximity ligation assay; ppITAM, double phosphorylated immune-receptor tyrosine activation motif; proxHCR, proximity-dependent hybridization chain reaction; PTK, protein tyrosine kinase; RCA, rolling circle amplification; SH2, Src Homology 2; Syk, spleen tyrosine kinase; WT, wild-type.

calibration and sophisticated data processing. In addition, data interpretation for superresolution microscopy can be tricky because of overcounting and underlabeling problems [5]. Traditional proximity-dependent assays to visualize protein interactions in living cells, such as bimolecular fluorescence complementation (BiFC) [6], Förster resonance energy transfer (FRET) [7]–based interaction assays including bioluminescence resonance energy transfer (BRET) [8], and proximity-dependent enzyme labeling techniques like ascorbic acid peroxidase (APEX) tagging [9], all rely on the expression of modified proteins and thus do not allow the study of the interaction and proximity of endogenous proteins. These can be monitored by an in situ proximity ligation assay (PLA) employing two oligo-coupled antibodies that, depending on their distance, direct the ligation of a DNA circle as a substrate for a nonlinear rolling circle amplification (RCA) process [10].

In contrast to RCA, the branched DNA (bDNA) method amplifies signals linearly through a sequential hybridization process [11]. This amplification procedure allows the detection of a single molecular target per cell with a broad detection range. The bDNA method has been extensively used in a microwell format to detect and quantify specific nucleic acid sequences, providing sensitive, specific, and reliable tools in the diagnosis of viral and bacterial infections [12]. More recently, bDNA has been combined with other technologies to allow the quantification of DNA or RNA targets in suspension cells or samples mounted on slides [13,14].

Based on the bDNA method, we developed a new branched proximity hybridization assay (bPHA), which allows one to monitor the proximity between target proteins by flow cytometer, thereby combining single-cell resolution with high throughput. We show that bPHA provides quantitative measurement of protein proximity in a large dynamic range. In addition, bPHA can be combined together with fluorescent staining in mixed cell populations. Employing bPHA, we studied the organization of the B cell antigen receptor (BCR), comprising a homodimeric membrane-bound immunoglobulin (mIg) molecule and the CD79a/CD79b heterodimer. We confirm the reorganization of different classes of the BCR upon B cell stimulation and demonstrate the potential of this new assay in cell signaling studies.

## Results

### The bPHA detects the dimeric organization of mIgM on the B cell surface

Similar to PLA, the bPHA employs a pair of oligos (plus and minus) that are coupled to target-binding probes such as nanobody, aptamer, or antibody (Fig 1A). If the two target proteins are proximal to each other, the specific binding probes are placing the two oligos close to each other so that they can serve as templates for a bDNA signal amplification [11]. Through a sequential hybridization to a pair of Z-DNA, preamplifier, amplifier, and finally, fluorescent label probes (Fig 1B–1D), theoretically, the bDNA method linearly amplifies the proximity signal 400 times [12,15]. Importantly, our design of bPHA does not use directly the proximity between the plus and minus oligos for signal amplification but, rather, through a pair of Z-DNA molecules as a bridge. Therefore, by simply changing one part of the Z-DNA sequence, a different set of preamplifier, amplifier, and fluorescent label probes can be used for bDNA signal amplification (Fig 1E). This allows one to switch the fluorescence of bPHA signals to adapt different experimental settings without making target protein-binding probes with another pair of oligos.

As a proof of concept, we employed bPHA to measure the proximity of BCR complexes on the surface of the human B cell line Ramos. The BCR on Ramos cells is a binding target of the DNA aptamer TD05 [16]. Secondary structure analysis predicts that the 5′ and 3′ ends of TD05 form a stable stem, which is unlikely to be involved in epitope binding (Fig 1F). Therefore, we generated two TD05 derivatives, one with a plus oligo attached to the 3′ end (TD05+) and another one with a minus oligo attached to the 5′ end (TD05−) of the TD05 (Fig 1F). We

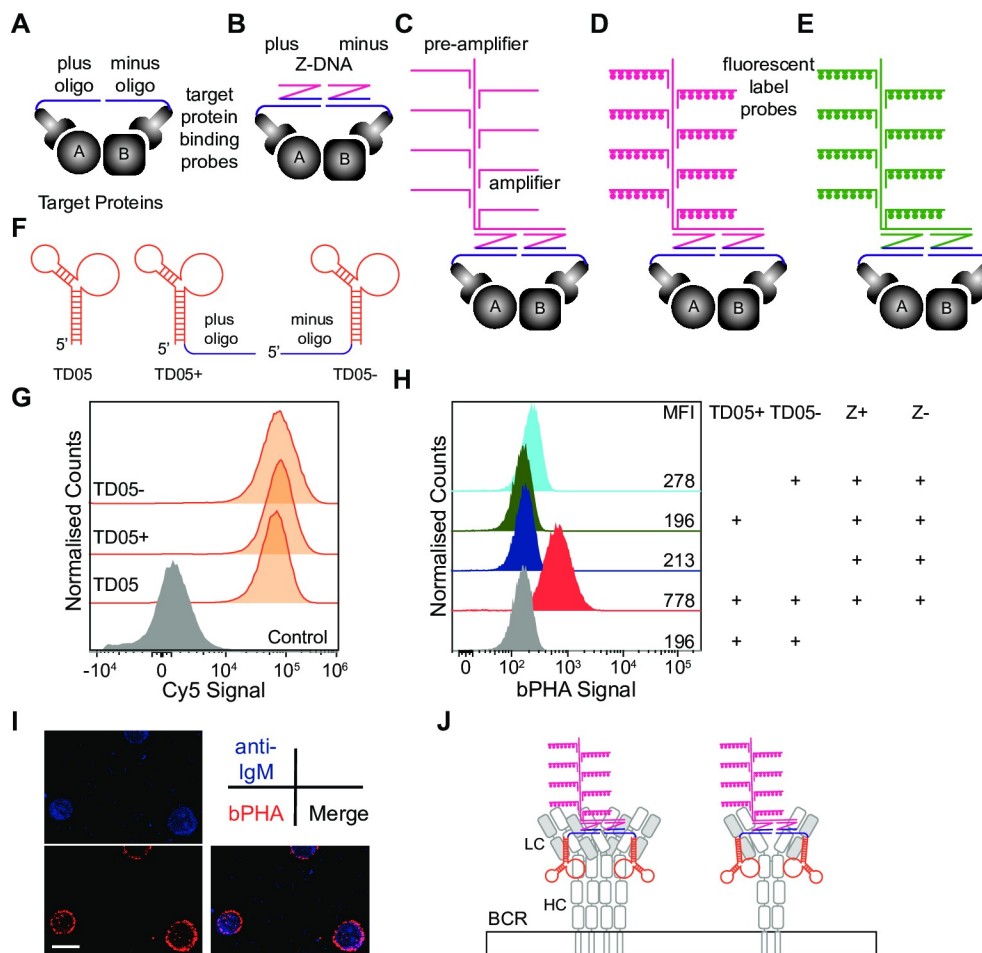

**Fig 1. The bPHA specifically detects the proximal localization of BCRs on Ramos cell surface.** (A-D) Schematic presentation of bPHA. The proximity of protein A and protein B is converted to the proximity of plus and minus oligos. The oligos are then hybridized to Z-DNA, followed by preamplifier, amplifier, and finally, fluorescent label probes. (E) Diagram showing that changing a part of the Z-DNA allows the change of fluorescence of bPHA signal. (F) Schematic presentation of TD05 aptamer and the TD05− and TD05+ derivatives. (G) Flow cytometry results showing similar staining of Ramos cells with Cy5-labeled TD05, TD05+, and TD05−. Cells stained with Cy5-coupled unrelated aptamer functioned as negative control. (H) TD05+:TD05− bPHA signal measured by flow cytometry for Ramos cells treated with the indicated probes. (I) Confocal microscopic images of Ramos cells after bPHA and anti-IgM staining. (G-I) Data represent at least three independent experiments. (J) Schematic drawing explaining what could be measured by the TD05+:TD05− bPHA. BCR, B cell antigen receptor; bPHA, branched proximity hybridization assay; Cy5, cyanine 5; HC, heavy chain; IgM, immunoglobulin M; LC, light chain; MFI, mean fluorescence intensity.

also generated cyanine 5 (Cy5)-coupled derivatives of TD05, TD05+, and TD05− and showed that an oligo extension of TD05 did not change the binding of the aptamer to the BCR on Ramos B cells (Fig 1G).

For bPHA, the Ramos cells were left untreated or exposed to saturating amounts of either TD05+, TD05−, or both aptamers. The cells were then washed, fixed, and incubated sequentially with Z-DNA and bDNA signal amplification components. Here, we used a pair of Z-DNA molecules that match to bDNA components generating fluorescence signal of Alexa Fluor 488. The bPHA signal was then quantified by flow cytometry with the setting for the green fluorescent protein/fluorescein isothiocyanate (GFP/FITC) channel (Fig 1H). Although cells treated with only the TD05− showed a weak increase of bPHA signal, a clear positive bPHA signal was only detected when the Ramos B cells were incubated with both the TD05

+ and TD05− aptamer and the complete bDNA components. Under the confocal microscope, the TD05+:TD05− bPHA signal was restricted to the Ramos cell surface, where it is colocalized with fluorescent anti-human IgM antibodies (Fig 1I). Taken together, these results suggest that the TD05+:TD05− bPHA detects a dimeric IgM complex (Fig 1J, left). However, we do not know the exact epitope covered by the TD05 aptamer on the mIgM molecule. The TD05 aptamer may bind once or twice to the homodimeric mIgM molecule (Fig 1J). In the latter case, the TD05+:TD05− bPHA signal may be specific for the mIgM homodimer (Fig 1J, right).

## The bPHA provides quantitative proximity measurements within a large dynamic range

To test the dynamic range of the bPHA signal, we expressed on Ramos B cells a GFP-μm fusion protein carrying an N-terminal GFP extension instead of the variable and the first constant domain (VH and CH1) of the μm heavy chain (Fig 2A). GFP is a specific binding target of the nanobody Enhancer (Enh) [17]. The used Enh construct carries a C-terminal LPETG sortag that can be site-specifically labeled with either a plus or a minus oligo through a sortase-mediated transpeptidation [18] (S1 Fig). We first used a Cy5-coupled Enh (Enh Cy5) to test for the GFP-μm expression on transfected Ramos cells and found that Cy5-Enh binding correlates well with GFP expression (Fig 2B). We then used the oligo-coupled Enh+ and Enh− for bPHA. Because it is necessary to distinguish the bPHA signal from the GFP signal, here we used another pair of Z-DNA molecules that match to bDNA components generating fluorescence signal of Alexa Fluor 647 and quantified the bPHA signal by flow cytometry with the setting for the Cy5/ allophycocyanin (APC) channel. A bPHA signal was only obtained with the GFP-positive Ramos B cells (Fig 2C). Interestingly, cells treated with Enh− alone did not give any bPHA signal compared with other controls (S2 Fig), suggesting that the weak signal observed previously in the cells treated with TD05− (Fig 1H) is likely to be experiment specific instead of a general problem of bPHA.

Importantly, the flow cytometry analysis of these cells shows that the bPHA signal is proportional to the GFP expression even at high fluorescence intensities (Fig 2D). When the single-cell fluorescence intensity values were exported for statistical analysis, the bPHA signal showed a linear correlation to the GFP signal with an $R^2$ value of 0.82, in spite of the fact that the GFP expression varied in a vast range (Fig 2E). Using plus and minus probe-coupled Enh, the proximity between the GFP domains was also measured by PLA. Similar to bPHA, positive PLA signal was only detected in the GFP-positive Ramos B cells (Fig 2F). However, in this GFP-positive population, the PLA signal is not proportional to the GFP expression level (Fig 2G). In fact, analysis of the exported single-cell fluorescence intensity values shows that the PLA signal is hardly correlated with the GFP signal, with an $R^2$ value of only 0.007 (Fig 2H).

These results thus demonstrated that unlike the PLA, the bPHA signal can be better read out by flow cytometry, and thus, bPHA provides an excellent quantitative measurement of protein proximity in a large dynamic range.

## The bPHA can quantify receptor proximity in a mixed cell population

Our analysis of the Ramos GFP-μm transfectants showed that bPHA signal detection might be combined with a flow cytometry analysis of other fluorescence signals, thus allowing a proximity analysis of a heterogenous cell population. As a proof of principle for this, we conducted the bPHA with a mixture of four different Ramos cell variants generated from the Ramos wild-type (WT) line by a CRISPR/Cas9-mediated gene knock-out and vector transfections (Fig 3A). The four Ramos variants expressed either the IgM-BCR or the IgD-BCR in the

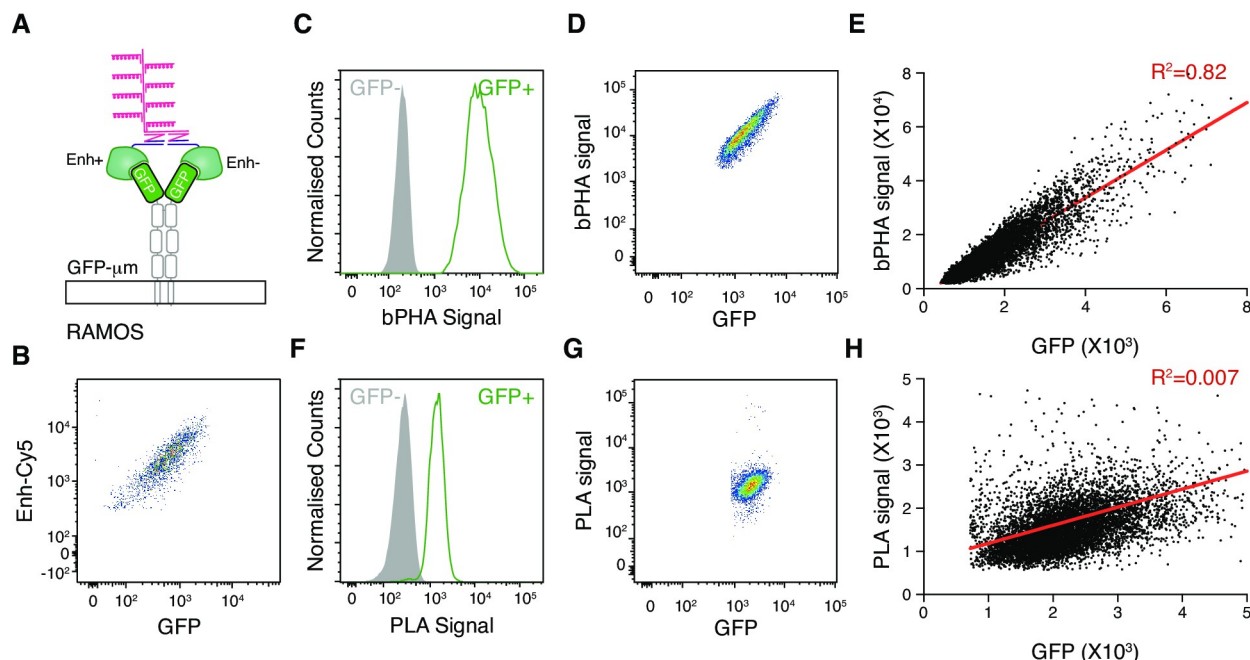

**Fig 2. The bPHA provides an excellent quantitative measurement of protein proximity in a large dynamic range.** (A) Schematic illustration showing that every GFP-μm expressed on Ramos cells surface have two GFPs in proximity that could be detected by Enh+:Enh− bPHA. (B) Flow cytometry results showing that the surface GFP-μm expression as assayed by Enh Cy5 staining is correlated with the GFP expression level. Gated for GFP+ cells. (C) Enh+:Enh− bPHA signal measured by flow cytometry for GFP− and GFP+ cells. (D) Flow cytometry results showing that the Enh+:Enh− bPHA signal is correlated with GFP signal. Gated for GFP+ cells. (E) Statistical analysis for data presented in (D). Raw data used for this plot are included in S1 Data. (F) Enh:Enh PLA signal measured by flow cytometry for GFP− and GFP+ cells. (G) Flow cytometry results showing that the Enh:Enh PLA signal is not correlated with GFP signal. Gated for GFP+ cells. (H) Statistical analysis for data presented in (G). Raw data used for this plot are included in S2 Data. Data are representative of a minimum of three independent experiments. bPHA, branched proximity hybridization assay; Cy5, cyanine 5; Enh, Enhancer; GFP, green fluorescent protein; PLA, proximity ligation assay.

presence or absence of the GFP-μm fusion protein and can be identified in a flow cytometry analysis by gating for IgD and GFP expression (Fig 3B). The amount of the BCR and the GFP-μm on the surface of the Ramos variant lines was determined by flow cytometry using TD05 Cy5 and Enh Cy5, respectively (S3 Fig). Note that the TD05 aptamer binds to both the IgM- and the IgD-BCR. The mixed Ramos cells were probed for BCR or GFP proximity by bPHA using TD05+/TD05− or Enh+/Enh− probe pairs and then stained with anti-human IgD phyco-erythrin (PE) antibody. The TD05+:TD05− and Enh+:Enh− bPHA signals were then quantified for each gated population by flow cytometry (Fig 3C and 3D). In agreement with the equal BCR expression on the surface of Ramos cell variants (S3A Fig), the quantification of the TD05+:TD05− bPHA signals gave similar values (Fig 3C). The Enh+:Enh− bPHA signal was only detected on the GFP+ cells and gave a higher value for the IgD-BCR-producing Ramos cells that also carry more GFP-μm fusion protein on their surface (compare Fig 3D with S3B Fig).

## A nanoscale receptor reorganization is detected on activated B cells by bPHA

The antigen-specific activation of mature B lymphocyte is accompanied by the opening of the oligomeric IgM-BCR and IgD-BCR and an increased IgM:IgD proximity [19]. To test for this nanoscale receptor reorganization, the IgD-BCR and GFP-μm-expressing Ramos B cells were directly exposed to saturate concentration of the oligo-coupled TD05 and/or Enh for 30 min

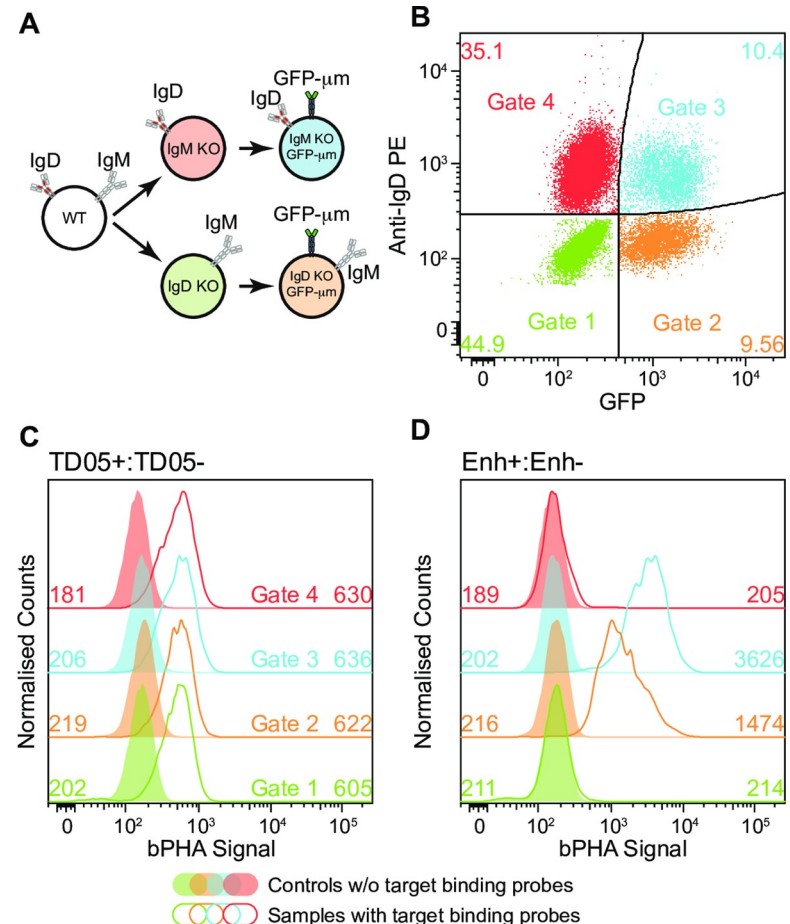

**Fig 3. The bPHA works in a mixed cell population.** (A) Schematic diagram showing the route map for generating GFP-μm-expressing IgM or IgD KO cells from WT Ramos B cells by CRSPR/Cas9 method. (B) Flow cytometry results showing that the subpopulations of the mixed cells can be identified by gating for the expression of GFP and anti-IgD staining. (C and D) TD05+:TD05− (C) or Enh+:Enh− (D) bPHA results were measured by flow cytometry and analyzed by the gating strategy shown in (B). Data represent three independent experiments. bPHA, branched proximity hybridization assay; Enh, Enhancer; GFP, Green fluorescent protein; IgD, immunoglobulin D; IgM, immunoglobulin M; KO, knock-out; PE, phycoerythrin; WT, wild-type.

for stimulation. The stimulated Ramos cells and untreated resting cells were then fixed and subjected to the TD05+:TD05−, TD05+:Enh−, and Enh+:Enh− bPHA analysis (Fig 4A). A weak TD05+:Enh− bPHA signal was detected on the resting IgD-BCR/GFP-μm Ramos cells, whereas this signal increased substantially upon B cell activation. Thus, an activation-dependent increased proximity of the IgD-BCR and GFP-μm proteins is detected on the Ramos B cell surface by bPHA (Fig 4B). A positive TD05+:TD05− bPHA signal was already monitored on resting Ramos cells and only slightly reduced upon cell stimulation, whereas the positive Enh+:Enh− bPHA signal remained unchanged on resting and stimulated Ramos cells (Fig 4A). This is in line with the equal staining of these cells with the TD05-Cy3 and Enh Cy5 reagents before and after stimulation (S4 Fig). Together, these results suggest that the TD05+:TD05− and Enh+:Enh− bPHAs mostly monitor the interdomain organization of the IgD-BCR and the GFP-μm chimeric molecule, respectively. Therefore, TD05 seems to have two identical binding sites inside the dimeric mIg molecule (Fig 4B).

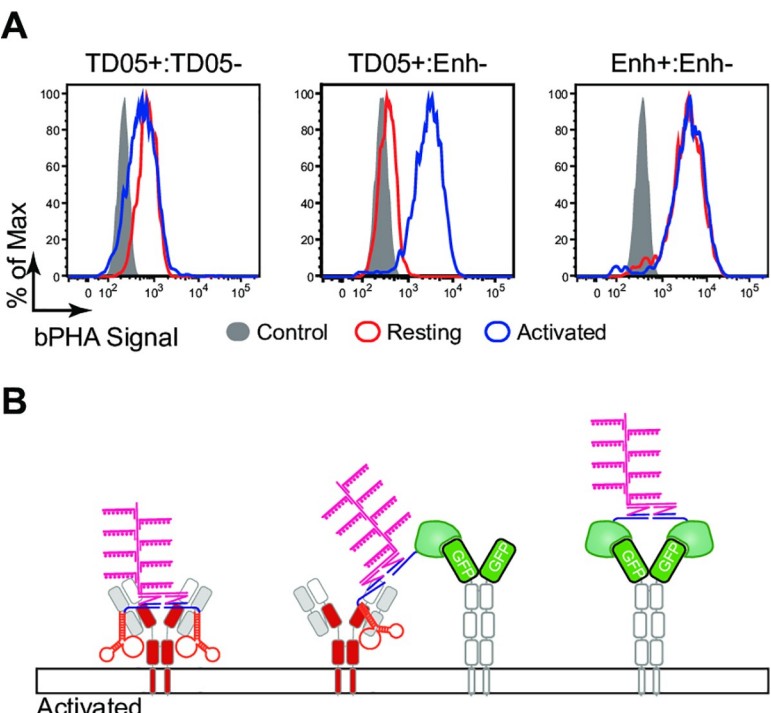

**Fig 4. The bPHA confirms the rearrangement of BCR upon stimulation.** (A) TD05+:TD05−, TD05+:Enh−, and Enh+:Enh− bPHA signals were measured by flow cytometry for resting and stimulated IgM-KO GFP-μm-expressing Ramos cells. The stimulated cells without the corresponding target binding probes served as control. Data represent four independent experiments. (B) Schematic diagrams showing that on the surface of IgM-KO GFP-μm-expressing Ramos cells surface, upon stimulation, IgD-BCR and GFP-μm mix together, producing positive TD05+:Enh− bPHA signal. BCR, B cell antigen receptor; bPHA, branched proximity hybridization assay; Enh, Enhancer; IgD, immunoglobulin D; IgM, immunoglobulin M; KO, knock-out.

## Class-specific kinetics of spleen tyrosine kinase recruitment to the activated BCR is monitored by bPHA

B cell activation is accompanied by the activation of several protein tyrosine kinases (PTKs) and an increase in PTK substrate phosphorylation. In particular, the spleen tyrosine kinase (Syk) plays a central role in the opening of the BCR and BCR signal amplification [20]. This involves the phosphorylation of the two tyrosines located in the immunoreceptor tyrosine-based activation motif (ITAM) of the BCR signaling subunits CD79a and CD79b and the binding of the tandem Src Homology 2 (SH2) domains of Syk to the double phosphorylated immune-receptor tyrosine activation motif (ppITAM) sequence [21,22]. Thus, the recruitment of Syk to the CD79a/CD79b heterodimer is a good indicator for BCR activation.

To monitor the increased Syk/BCR proximity during the Syk recruitment process on a single-cell basis, we developed an intracellular bPHA (Fig 5A). For this, we first labeled anti-Syk and anti-CD79a antibodies with the minus and plus oligo, respectively, using a cross-linker and click chemistry. The labeling efficiency of the antibodies was verified by SDS-PAGE (S5 Fig). Enriched splenic B cells from C57BL/6 mice were left resting or stimulated for different times (1, 5, 10 min) with either anti-IgD or anti-IgM antibodies. After fixation and permeabilization of the cells, we employed the oligo-coupled anti-CD79a and anti-Syk antibodies for bPHA (Fig 5B). An anti-CD79a+:anti-Syk− bPHA signal was already detected in some of the resting B cells. However, this signal increased in the majority of B cells upon their stimulation,

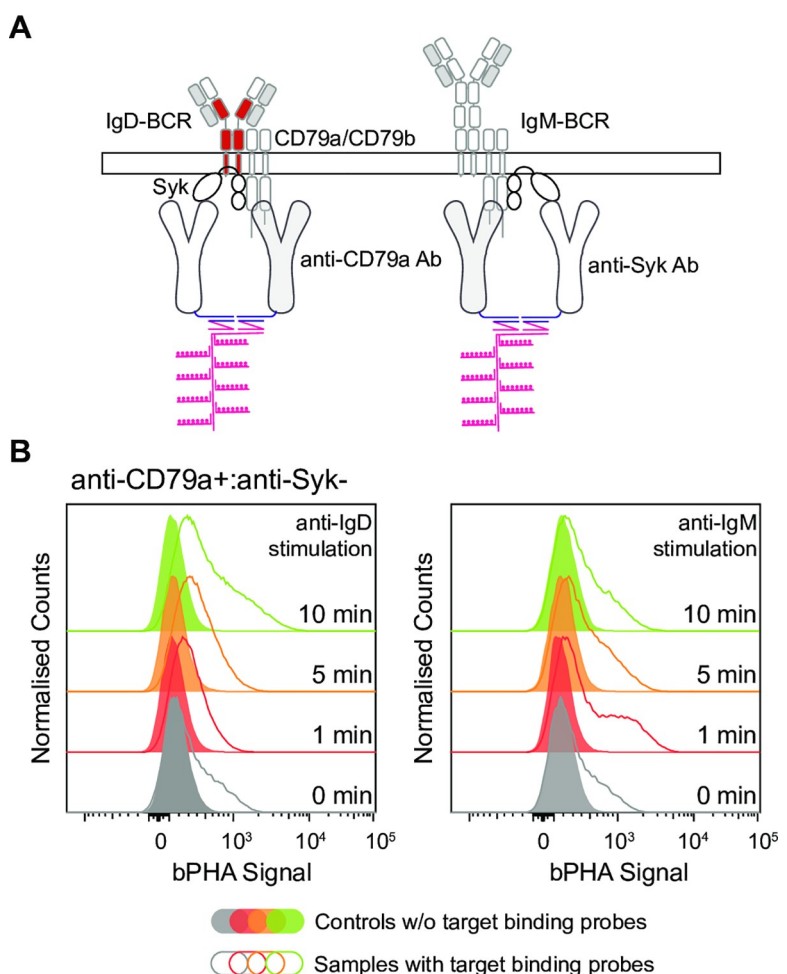

**Fig 5. The bPHA uncovers class-specific kinetics of Syk recruitment to BCR.** (A) Schematic diagrams showing that upon stimulation, Syk can be recruited to CD79a. The proximity between CD79a and Syk can be measured by bPHA using oligo-coupled anti-CD79a and anti-Syk antibodies. (B) Anti-CD79a+:anti-Syk− bPHA signals were measured by flow cytometry for resting and anti-IgD- or anti-IgM-stimulated splenic B cells. Cells without the corresponding target binding probes served as controls for both the resting and stimulated cells. Data represent three independent experiments. BCR, B cell antigen receptor; bPHA, branched proximity hybridization assay; IgD, immunoglobulin D; IgM, immunoglobulin M; Syk, spleen tyrosine kinase.

indicating a recruitment of Syk to the BCR. Interestingly, our intracellular bPHA depicted class-specific differences in the recruitment of Syk to the BCR. In the anti-IgM-stimulated B cells, the anti-CD79a+:anti-Syk− bPHA signal peaked at 1 min, whereas in the anti-IgD-stimulated B cells, the signal was still increasing 10 min after the stimulation. This is consistent with a previous western blot study showing that IgD-BCR induces a stronger and more prolonged protein tyrosine phosphorylation than IgM-BCR upon stimulation [23]. Unlike the western blot, however, the bPHA offers single-cell resolution and delineates heterogeneity in the Syk recruitment of the activated B cell population. Depending on the time point, only 30%–50% of the B cells show an increased Syk:CD79a proximity, indicating that BCR signaling is not synchronized in all B cells, probably because of the heterogeneity of surface BCR expression in the isolated splenic B cells (S6 Fig).

## Discussion

We here demonstrated that the bDNA signal amplification method can be adopted for a new proximity assay with nanometer resolution. Most cellular processes depend on a close proximity of molecular components, but these nanoscale organizations are difficult to study with the light microscope. Thus, in recent years, several new proximity methods have been developed [24]. Among them, PLA offers a unique advantage to study the organization of unmodified endogenous proteins. PLA was originally designed to visualize protein–protein interactions in situ. PLA results are captured as microscopic images, and PLA dots are counted from these images [10]. As PLA requires imaging processing, its throughput is limited to a few hundred cells per experiment. Recently, PLA was combined with flow cytometry to simplify data acquisition and improve throughput [25–27]. However, because of the fact that the signaling amplification of PLA through RCA is nonlinear, the PLA signals are bound to saturate at a higher density [28]. In our hands, staining cells with high amounts of probes often generates indistinguishable saturated PLA signals for samples and controls, whereas staining cells with low amounts of probes produces weak signals that are hardly detected by flow cytometer. We thus prefer to quantify PLA results by counting dots of microscopic images [20,29,30].

Except for enzyme-catalyzed amplification, an oligonucleotide can also be amplified by hybridization. Recently, the proximity-dependent hybridization chain reaction (proxHCR) method was described for detecting protein proximities in microscopy and flow cytometry using a local HCR to amplify signals from two closely positioned hairpin oligonucleotides [31]. Because it also depends on the dual binding of proximity probes, proxHCR reaches similar specificities to in situ PLA. However, it was not determined whether the HCR method is able to amplify the proximity signal linearly.

With bPHA, the protein proximity signal is linearly amplified by bDNA that has a broad detection range over four to five orders of magnitude [12]. Thus, this amplification procedure can be combined with flow cytometry, thereby achieving high throughput (thousands to millions of cells per measurement) with single-cell resolution. As proof of principle, we showed in our flow cytometric analysis of dimeric GFP-μm–expressing Ramos B cells that the bPHA signals are linearly correlated with GFP expression over a big range. In addition, modern flow cytometers are equipped with multiple lasers, enabling bPHA measurement in mixed cell populations, as we have shown in this work. By using aptamers or nanobodies instead of Fab fragments or whole antibodies, we also improved the molecular resolution of the proximity assay. A limitation of this high-resolution assay is that, in the case of dimeric proteins, our bPHA preferentially detects intra- instead of intermolecular proximity.

Similar to PLA, for bPHA, the proximity between two target proteins is translated via oligo-coupled binding reagents into the proximity of a plus and minus oligo. However, in our design for bPHA, the proximity between the plus and minus oligos is not directly used for bDNA signal amplification but, rather, for the close pairing of two Z-DNAs. As we have shown here, this design allows us to switch between bDNA components with different fluorescence dyes to fit with different conditions using the same pair of oligos. Furthermore, different molecules in the same cell could be labeled with their specific binding probes carrying different oligo sequences. The proximities between these different oligos might then be picked up and amplified by different pairs of Z-DNA and amplifier reagents, thus making it appropriate to multiplex bPHA reactions for monitoring an array of molecular proximities.

By using nonpermeabilized cells, the bPHA can specifically monitor the proximity of molecules on the cell surface. In this way, we show here that the IgD-BCR and IgM-BCR are segregated from each other on resting B cells but concatenated after B cell stimulation, thus confirming previous Fab-PLA and superresolution microscopy studies [19]. After a cellular

 

permeabilization, our bPHA method can, however, also monitor intracellular molecular proximities. This we employed for a kinetic study of the Syk recruitment to the BCR in activated splenic B murine cells. We found that anti-IgD stimulation results in a more prolonged and prominent Syk recruitment than the anti-IgM stimulation, which is in line with the kinetics of protein phosphorylation upon IgD-BCR and IgM-BCR activation [23].

## Materials and methods

### Ethics statement

The animals were maintained and used in agreement with The German Animal Welfare Act (Das deutsche Tierschutzgesetz). Animals were sacrificed under the project X-17/10C and registered with the Committee on Animal Experimentation of the Regional Council Freiburg, Germany (Tierversuchskommission des Regierungspräsidiums Freiburg, Deutschland).

### Binding reagents

For surface staining of Ramos cells, the following anti-human antibodies, aptamers, and nanobodies were used: anti-IgM eFluor 450 (SA-DA4, eBioscience, 12-9998-42), TD05 Cy5, TD05+ Cy5, TD05− Cy5, TD05 Cy3 (custom order from Sigma-Aldrich), Enh Cy5 (homemade), and anti-IgD PE (IA6-2, BD Bioscience, 555779). For surface staining of mouse splenic B cells, the following anti-mouse antibodies were used: anti-IgM eFluor 450 (eB121-15F9, eBioscience, 14-5890-82) and anti-IgD FITC (11-26c.2a, Biolegend, 405704). For stimulating mouse splenic B cells, anti-IgM (polyclonal, SouthernBiotech, 2020–01) and anti-IgD (polyclonal, eBioscience, 24-5093-51) were used. For bPHA probe preparation, the following antibodies were used: anti-CD79a (JCB117+HM47/A9, abcam, ab213114) and anti-Syk (SYK-01, Biolegend, 626202).

### Cell culture

All Ramos cell lines (German Collection of Microorganisms and Cell Cultures DSMZ ACC603) were cultured in RPMI medium (Gibco, with stable glutamine) supplemented with 10% FCS (Biochrom), 10 units/mL penicillin/streptomycin (Gibco), and 50 mM beta-mercaptoethanol (Sigma-Aldrich) at 37˚C in 5% $CO_2$. The Phoenix cell line was cultured in Iscove's medium (Biochrom, with stable glutamine) supplemented with 10% FCS (PAN Biotech), 10 units/mL penicillin/streptomycin (Gibco), and 50 mM beta-mercaptoethanol (Sigma-Aldrich) at 37˚C in 7.5% $CO_2$.

### Retroviral transduction

Retroviral transductions in Ramos cells were performed as previously described [32]. In brief, Phoenix cells were transfected using PolyJet DNA in vitro transfection reagent following the manufacturer's protocol (SignaGen Laboratories). Retrovirus-containing supernatants were collected 48 h after transfection and used for transduction.

### Isolation of mouse splenic B cells

Total splenocytes were isolated from 8-wk-old to 12-wk-old C57BL/6 mice (female or male). Splenic B cells were enriched by MACS depletion of CD43+ cells using anti-CD43 magnetic beads (Miltenyi Biotech) according to the manufacturer's instructions. The purified splenic B cells were cultured for a minimum of 2 h at 37˚C in 7.5% $CO_2$ in Iscove's medium (Biochrom, with stable glutamine) supplemented with 10 units/mL penicillin/streptomycin, 50 mM beta-mercaptoethanol (Sigma-Aldrich), and 10% FCS (PAN Biotech). The splenic B cells were then

further analyzed for IgM- and IgD-BCR surface expression or stimulated with antibodies, and the recruitment of Syk to CD79a by bPHA was detected.

### Flow cytometry analysis

For Ramos cells, $2 \times 10^5$ cells were stained with TD05 Cy5 (1 mM), TD05+ Cy5 (1 mM), TD05 Cy3 (1 mM), TD05− Cy5 (1 mM), Enh Cy5 (1.5 μg/mL), or anti-IgD PE (1:100) in 100 μL of PBS. For mouse splenic B cells, $2 \times 10^5$ cells were stained with anti-IgM eFluor 450 (1:100) and anti-IgD FITC (1:100) in 100 μL of PBS. Staining was performed at 4°C for 15 min. After washing, samples were measured with the Attune NxT Flow Cytometer (Thermo Fisher Scientific). Data were exported in FCS-3.0 format and analyzed with FlowJo software (TreeStar).

### Confocal imaging

After bPHA and staining, cells were transferred to 18-well μ-slides (ibidi) and rested for 15 min to allow the attachment of cells to the slides. Samples were then imaged using a Zeiss 780 Meta confocal microscope (Carl Zeiss) equipped with a Zeiss Plan-Apochromat 63× oil immersion objective lens. Images were processed using ImageJ.

### Sortase-mediated transpeptidation

The pentamutant sortase [33] was expressed in *Escherichia coli* with a 6xHis tag at the C terminus and purified by Ni-NTA. The sortase-mediated transpeptidation was performed overnight at 4°C in 50 mM Tris (pH 7.5), 150 mM NaCl, and 10 mM $CaCl_2$ sortagging buffer by mixing 100 μM Enh with 500 μM GGG-oligo (plus oligo: TGCATAATCACCACTAAAACTGTAAA GCT AAGTGA or minus oligo: GTTACGAAACACGCTCTAAGTCTCTAAACTCGAAT, ordered from Biomers) and 2.5 μM sortase. Afterward, the His-tagged sortase and remaining His-tagged, unlabeled Enh and His-tagged Gly residue produced during sortagging were all removed by passing over a Ni-NTA column (Qiagen).

### SDS-PAGE

Protein samples were mixed with 5× nonreducing/reducing loading buffer and then heated at 95°C for 5–10 min. Protein marker (PageRule Prestained 10–180 kDa Protein Ladder, Thermo Fisher Scientific) and equal amounts of proteins were loaded and separated on 12.5% Tris-glycine SDS-PAGE gels. Gels were stained in 20–30 mL protein staining solution (Instant BlueTM, expedeon) overnight. The next day, gels were imaged by Molecular Imager Gel DocTM XR+ (BioRad). All recorded images were analyzed with Image Lab software.

### Antibody labeling

To label antibodies with oligo, 100 μg (0.67 nmole) of anti-CD79a and anti-Syk were first mixed with 20 nmole cross-linker DBCO-Sulfo-NHS-ester (762040, Sigma-Aldrich). Samples were incubated at 37°C for 60 min. After desalting (Zeba spin desalting columns, Thermo Fisher Scientific), cross-linker-activated antibodies were mixed with 12 nmole of either plus or minus oligos (Azid-PEG4 modified at 5′ for the plus and 3′ for the minus oligo, ordered from Biomers). Samples were then kept at 37°C for 30 min. Labeled antibodies were kept at 4°C.

### bPHA

For measuring the proximity between BCRs (TD05+:TD05−), between GFP domains of GFP-μm (Enh+:Enh−), or between BCR and GFP-μm (TD05+:Enh−) by bPHA, $1 \times 10^6$ Ramos WT or mutant cells were aliquoted and washed with DPBS (Sigma-Aldrich). Cells were stained in

100 μL of DPBS with the corresponding oligo-coupled TD05 and/or Enh probes at 4°C for 30 min and fixed with the PrimeFlow fixation buffer 1 (PrimeFlow RNA Assay, Thermo Fisher Scientific) in the dark for 30 min at 4°C.

For detecting the reorganization of BCR upon stimulation, cells first fixed and later stained with bPHA probes were treated as resting cells, whereas cells stained with bPHA probes for 30 min at 4°C and then fixed were treated as stimulated cells.

To monitor the recruitment of Syk to CD79a, $2.5 \times 10^6$ mouse splenic B cells were aliquoted, washed with DPBS (Sigma-Aldrich), resuspended in 500 μL DPBS, and cultured at 37°C for 20–30 min. Cells were stimulated with anti-mouse-IgM (1:500) or anti-mouse-IgD (1:500) for 1, 5, and 10 min, respectively. Untreated cells were used as 0-min control. After fixation, cells were permeabilized using the PrimeFlow Permeabilization Buffer (PrimeFlow RNA Assay, Thermo Fisher Scientific), stained with anti-CD79a plus and anti-Syk minus probes at 4°C for 30 min, and then fixed again with the PrimeFlow fixation buffer 2 (Prime-Flow RNA Assay, Thermo Fisher Scientific) in the dark for 60 min at room temperature (RT).

The bPHA probe final concentration was as follows: TD05+, 1 μM; TD05−, 1 μM; Enh+, 1.5 μg/mL; Enh−, 1.5 μg/mL; anti-CD79a+, 5 μg/mL; anti-Syk−, 0.25 μg/mL.

Cells already labeled with the corresponding bPHA probes and fixed then underwent bDNA signal amplification according to the manufacturer's instructions of the PrimeFlow RNA Assay Kit (Thermo Fisher Scientific). In brief, cells were first hybridized with the Z-DNA pairs (1:20 in target probe diluent) for 2 h at 40°C. After washing with the wash buffer (2×, 5 min, RT), bPHA signals were further amplified and fluorescently labeled by a sequential hybridization with the standard PreAmp Mix (1.5 h), Amp Mix (1.5 h), and Label Probes mix (1:200 in target probe diluent, 1 h) at 40°C with 2×, 5-min RT wash with the wash buffer after each hybridization step.

For measuring protein proximity in a mixed cell population, cells were further stained with anti-human IgD PE (1:100) for 15 min at 4°C.

bPHA signals were then measured by Attune NxT Flow Cytometer (Thermo Fisher Scientific).

## Flow PLA

Purified Enh was coupled to Duolink PLA probes following the manufacturer's instructions of Duolink PLA Probemaker PLUS and Duolink PLA Probemaker MINUS kits (Sigma-Aldrich). The flow PLA is performed based on the Duolink PLA Flow Cytometry protocol (Sigma-Aldrich) with modification using the Duolink flowPLA Detection Kit–Orange (Sigma-Aldrich). In brief, $5 \times 10^5$ cells were washed with PBS and then incubated with plus and minus oligo-labeled Enh (1:5 each in Duolink Probe Diluent) for 1 h at 37°C. After washing twice with ice-cold PBS, the cells were fixed by 2% paraformaldehyde (PFA) at RT for 20 min and then blocked with Duolink Blocking Solution (1 h, 37°C) after washing. Washing with PBS again, ligase (1:40 in Duolink Ligation buffer) was then mixed with the cells and incubated for 30 min at 37°C for ligation to happen. After washing twice with Duolink Wash Buffer A, the cells were then incubated with polymerase (1:80 in amplification buffer) overnight at 37°C. After washing twice with Duolink Wash Buffer B, the cells were incubated for 30 min at 37°C with 1X flowPLA Detection Solution. After washing twice more with Duolink Wash Buffer B, the cells were then subjected to measurement using Attune NxT Flow Cytometer (Thermo Fisher Scientific).

## Supporting information

**S1 Fig. Sortase-mediated site-specific labeling of Enh with oligo extensions for bPHA.** (A) Schematic presentation of the labeling reaction. (B and C) Coomassie-stained 12.5% reducing

SDS-PAGE gel showing the composition of materials after sortase-mediated transpeptidation (B) or after further clearance with Ni-NTA column (C). bPHA, branched proximity hybridization assay; Enh, Enhancer; Ni-NTA; nickel-nitrilotriacetic acid.
(TIF)

**S2 Fig. Enh+:Enh− bPHA signal measured by flow cytometry for GFP-μm-expressing Ramos cells treated with the indicated probes.** bPHA, branched proximity hybridization assay; Enh, Enhancer; GFP; green fluorescent protein.
(TIF)

**S3 Fig. Surface BCR and GFP-μm levels in mixed cell population.** (A and B) Flow cytometry results showing the surface IgD- or IgM-BCR level evaluated by TD05 Cy5 staining (A) or GFP-μm level by Enh Cy5 staining (B) for the mixed Ramos cells following the gating strategy shown in Fig 3B. BCR, B cell antigen receptor; Cy5, cyanine 5; Enh, Enhancer; GFP, green fluorescent protein; IgD, immunoglobulin D; IgM, immunoglobulin M.
(TIF)

**S4 Fig. Surface IgD-BCR and GFP-μm levels are not changed upon stimulation.** (A and B) Flow cytometry results showing the surface IgD-BCR level evaluated by TD05 Cy3 staining (A) or GFP-μm level by Enh Cy5 staining (B) for the resting and activated IgM-KO GFP-μm-expressing Ramos cells. BCR, B cell antigen receptor; Cy3, cyanine 3; Cy5, cyanine 5; Enh, Enhancer; GFP, green fluorescent protein; IgD, immunoglobulin D; IgM, immunoglobulin M; KO, knock-out.
(TIF)

**S5 Fig. The 12.5% reducing TGX Stain-Free gel showing the composition of antibodies after coupling to the oligo extensions.** TGX; tris-glycine extended.
(TIF)

**S6 Fig. Flow cytometry results showing the heterogeneity of mouse splenic B cells in terms of the surface expression of IgD- and IgM-BCR.** BCR, B cell antigen receptor; IgD, immunoglobulin D; IgM, immunoglobulin M.
(TIF)

**S1 Data. Excel spreadsheet containing the underlying numerical data for Fig 2E.**
(XLSX)

**S2 Data. Excel spreadsheet containing the underlying numerical data for Fig 2H.**
(XLSX)

**S1 Raw Images. Raw images of S1B Fig, S1C Fig, and S5 Fig.**
(PDF)

## Acknowledgments

We thank Dr. Lise Leclercq for critical reading of this manuscript. We thank Dr. Palash C. Maity for the IgM- and IgD-BCR knock-out of Ramos cells. We thank Dr. Julia Jellusova for her help in experimental mice.

## Author Contributions

**Conceptualization:** Michael Reth, Marco Cavallari, Jianying Yang.

**Data curation:** Shuangshuang Zheng, Melanie Sieder, Michael Mitterer.

**Formal analysis:** Shuangshuang Zheng, Melanie Sieder, Jianying Yang.

**Funding acquisition:** Michael Reth, Jianying Yang.

**Investigation:** Shuangshuang Zheng, Michael Mitterer, Jianying Yang.

**Methodology:** Shuangshuang Zheng, Melanie Sieder, Michael Mitterer, Marco Cavallari, Jianying Yang.

**Project administration:** Jianying Yang.

**Resources:** Marco Cavallari.

**Supervision:** Michael Reth, Jianying Yang.

**Validation:** Shuangshuang Zheng, Melanie Sieder, Michael Mitterer.

**Visualization:** Shuangshuang Zheng, Melanie Sieder, Michael Mitterer, Marco Cavallari, Jianying Yang.

**Writing – original draft:** Shuangshuang Zheng, Marco Cavallari, Jianying Yang.

**Writing – review & editing:** Michael Reth, Jianying Yang.

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
