## [Editor Report · Decision Letter 0]

27 Jun 2019

Dear Dr YANG, 

Thank you for submitting your manuscript entitled "A new branched proximity hybridization assay for the quantification of nanoscale protein-protein proximity" for consideration as a Methods and Resources by PLOS Biology.

Your manuscript has now been evaluated by the PLOS Biology editorial staff as well as by an academic editor with relevant expertise and I am writing to let you know that we would like to send your submission out for external peer review.

**Important**: Please also see below for further information regarding completing the MDAR reporting checklist. The checklist can be accessed here: https://plos.io/MDARChecklist

Please re-submit your manuscript and the checklist, within two working days, i.e. by Jun 29 2019 11:59PM.

Kind regards,

Di Jiang, PhD

Associate Editor

PLOS Biology

INFORMATION REGARDING THE REPORTING CHECKLIST:

PLOS Biology is pleased to support the "minimum reporting standards in the life sciences" initiative (https://osf.io/preprints/metaarxiv/9sm4x/). This effort brings together a number of leading journals and reproducibility experts to develop minimum expectations for reporting information about Materials (including data and code), Design, Analysis and Reporting (MDAR) in published papers. We believe broad alignment on these standards will be to the benefit of authors, reviewers, journals and the wider research community and will help drive better practise in publishing reproducible research. 

We are therefore participating in a community pilot involving a small number of life science journals to test the MDAR checklist. The checklist is intended to help authors, reviewers and editors adopt and implement the minimum reporting framework. 

IMPORTANT: We have chosen your manuscript to participate in this trial. The relevant documents can be located here:

MDAR reporting checklist (to be filled in by you): https://plos.io/MDARChecklist

**We strongly encourage you to complete the MDAR reporting checklist and return it to us with your full submission, as described above. We would also be very grateful if you could complete this author survey:

https://forms.gle/seEgCrDtM6GLKFGQA

Additional background information:

Interpreting the MDAR Framework: https://plos.io/MDARFramework

Please note that your completed checklist and survey will be shared with the minimum reporting standards working group. However, the working group will not be provided with access to the manuscript or any other confidential information including author identities, manuscript titles or abstracts. Feedback from this process will be used to consider next steps, which might include revisions to the content of the checklist. Data and materials from this initial trial will be publicly shared in September 2019. Data will only be provided in aggregate form and will not be parsed by individual article or by journal, so as to respect the confidentiality of responses. 

Please treat the checklist and elaboration as confidential as public release is planned for September 2019.

We would be grateful for any feedback you may have.

---

## [Decision Letter · Decision Letter 1]

8 Aug 2019

Dear Dr YANG,

Thank you very much for submitting your manuscript "A new branched proximity hybridization assay for the quantification of nanoscale protein-protein proximity" for consideration as a Methods and Resources at PLOS Biology. Your manuscript has been evaluated by the PLOS Biology editors, an Academic Editor with relevant expertise, and by two independent reviewers.

In light of the reviews (below), we will not be able to accept the current version of the manuscript, but we would welcome resubmission of a revised version that takes into account the reviewers' comments. You will need to include improved comparisons with other methods as requested by reviewer 1. In addition, our Academic Editor encourages you to show that the ability to quantify the interaction can be used to learn new biology, but this isn't mandatory. We cannot make any decision about publication until we have seen the revised manuscript and your response to the reviewers' comments. Your revised manuscript is also likely to be sent for further evaluation by the reviewers.

Your revisions should address the specific points made by each reviewer. Please submit a file detailing your responses to the editorial requests and a point-by-point response to all of the reviewers' comments that indicates the changes you have made to the manuscript. In addition to a clean copy of the manuscript, please upload a 'track-changes' version of your manuscript that specifies the edits made. This should be uploaded as a "Related" file type. You should also cite any additional relevant literature that has been published since the original submission and mention any additional citations in your response. 

Before you revise your manuscript, please review the following PLOS policy and formatting requirements checklist PDF: http://journals.plos.org/plosbiology/s/file?id=9411/plos-biology-formatting-checklist.pdf. It is helpful if you format your revision according to our requirements - should your paper subsequently be accepted, this will save time at the acceptance stage.

Please note that as a condition of publication PLOS' data policy (http://journals.plos.org/plosbiology/s/data-availability) requires that you make available all data used to draw the conclusions arrived at in your manuscript. If you have not already done so, you must include any data used in your manuscript either in appropriate repositories, within the body of the manuscript, or as supporting information (N.B. this includes any numerical values that were used to generate graphs, histograms etc.). For an example see here: http://www.plosbiology.org/article/info%3Adoi%2F10.1371%2Fjournal.pbio.1001908#s5.

For manuscripts submitted on or after 1st July 2019, we require the original, uncropped and minimally adjusted images supporting all blot and gel results reported in an article's figures or Supporting Information files. We will require these files before a manuscript can be accepted so please prepare them now, if you have not already uploaded them. Please carefully read our guidelines for how to prepare and upload this data: https://journals.plos.org/plosbiology/s/figures#loc-blot-and-gel-reporting-requirements.

Upon resubmission, the editors will assess your revision and if the editors and Academic Editor feel that the revised manuscript remains appropriate for the journal, we will send the manuscript for re-review. We aim to consult the same Academic Editor and reviewers for revised manuscripts but may consult others if needed.

We expect to receive your revised manuscript within two months. Please email us (plosbiology@plos.org) to discuss this if you have any questions or concerns, or would like to request an extension. At this stage, your manuscript remains formally under active consideration at our journal; please notify us by email if you do not wish to submit a revision and instead wish to pursue publication elsewhere, so that we may end consideration of the manuscript at PLOS Biology.

When you are ready to submit a revised version of your manuscript, please go to https://www.editorialmanager.com/pbiology/ and log in as an Author. Click the link labelled 'Submissions Needing Revision' where you will find your submission record. 

Sincerely,

Di Jiang, PhD

Associate Editor

PLOS Biology

Reviewer remarks:

Reviewer #1: The paper “A new branched proximity hybridization assay for the quantification of nanoscale protein-protein proximity” by Zheng et al. describes an approach to demonstrate colocation of proteins directly in biological samples. It is a matter of increasing importance to understand functional properties of cellular systems by observing how proteins may be interacting to exert their activities, and improved methods for their analysis can be of value.

The approach the authors describe is a variant of the widely used in situ PLA technique, commercially available from SigmaAldrich, and applicable both in situ and for flow cytometry as also shown by Zheng et al. using their bHPA technique. In both bHPA and in situ PLA samples are first treated with pairs of oligonucleotide-conjugated protein-binding reagents, such as antibodies. Zheng et al. demonstrate proximal binding by a hybridization-based local amplification technique called branched DNA, previously reported in reference 11 and here used via a kit, commercially available from ThermoFisher. By contrast, in the in situ PLA technique oligonucleotides on pairs of reagents brought in proximity serve as templates for ligation reactions that produce DNA circles. These DNA circles are then locally amplified via enzyme-catalyzed rolling circle amplification, resulting in prominent detection signals. 

Zhen et al. demonstrate the ability of their technique to reveal protein interactions in the context of signaling via the B cell antigen receptor on cell lines and mouse splenocytes, but the focus of the paper is on the method rather than the biology. I have some specific comments as follows:

1. The branched DNA technique depends on that signals should only be detected when both members of the Z-DNA pairs have bound in proximity and can contribute to the strength of hybridization. In that regard it is of some concern that the bPHA signals illustrated by the light blue peak in Figure 1h are somewhat elevated over the background (shifted to the right), although still considerably lower than the red peak that illustrates the situation when all required reagents were present. This indicates that the “TD05-“ reagent alone can elicit some signal, something that could complicate analysis of protein interactions when target proteins are evaluated across a wide range of expression levels. The authors should comment on this, or preferably find conditions where this background is not observed.

2. The reported 400-fold signal amplification (lines 16 and 68), should be supported either by referring to information about the kit or own measurements.

3. In the abstract the bHPA technique is said to offer better quantification of protein proximity (line 13), without mentioning what this is in comparison to. Elsewhere it is clear that the authors are comparing their approach to in situ PLA as is appropriate. Unfortunately, they do not provide any data from experimental side-by-side comparisons, however, making it difficult to evaluate to what extent their technique represents real progress. In situ PLA is said to result in a nonlinear amplification process (lines 42 and 194), but this claim is supported by reference 27, which represents a not fully relevant comparison between FRET and in situ PLA. On theoretical grounds it is not obvious why branched DNA amplification would yield a more linear result than rolling circle amplification, so such claims need to be supported by evidence from direct comparisons. 

4. The novel element of the paper is the use of branched DNA as an enzyme-free signal amplification method to reveal protein proximity via pairs of oligonucleotide-conjugated antibodies. It would therefore be relevant also to refer to and briefly discuss the paper “Proximity-dependent initiation of hybridization chain reaction” Bjorn Koos et al., 2015, DOI: 10.1038/ncomms8294. Like bHPA and in situ PLA this paper describes analysis of protein proximity via similar pairs of DNA-conjugated reagents but using another enzyme-independent local amplification technique namely a hybridization chain reaction. 

5. On lines 69 and 213 the claim is made that bHPA differs from in situ PLA in that in bHPA the oligonucleotides attached to the affinity reagents are not directly used for amplification, permitting multiplexing. This is a misunderstanding, however, as for in situ PLA secondarily added oligonucleotides are converted to a DNA circle used for amplification, and similar to bHPA this renders the assays suitable for multiplexing. (By contrast, in PLA used for protein detection in solution phase, the antibody-conjugated DNA strands are directly amplified via PCR)

6. Since the purpose of the paper is to demonstrate the advantages of a local amplification method different from that used in the established in situ PLA technique for protein proximity analyses, it would be preferable if side-by-side experimental comparisons were presented to help the reader evaluate how the two procedures compare. It would also be helpful to offer a theoretical comparison, perhaps in a table form, between bHPA, in situ PLA and the above mentioned proximity-dependent hybridization chain reaction. For example, the latter and bHPA differ from in situ PLA in that no enzyme is required for signal amplification, potentially simplifying automation.

Reviewer #2: In the manuscript entitled ” A new branched proximity hybridization assay for the quantification of nanoscale protein-protein proximity” Zheng et al describes a method to probe for proximity, using branched hybridization to obtain a signal amplification. The authors show convincing data on the performance of the method. The advantage with the methods compared with methods such as PLA is that it does not require any enzymatic steps, hence should be more inexpensive. Novel methods for biology are needed and the development described herein will be of interest to the scientific community. I only have a minor suggestions to the authors.

1. The method section is rather short and it would be valuable if more details could be added, e.g. sequences of oligos used, buffer conditions for hybridization, where was sortase obtained? For a reader who want to replicate the experiments or use the method, it is important to provide all details regarding the method.

2. Why the intermediate step with hybridization of Z-DNA, can the pre-amplifier be hybridized directly to the minus and plus probe? That would remove one step of the assay.

---

## [Decision Letter · Decision Letter 2]

31 Oct 2019

Dear Dr YANG,

Thank you for submitting your revised Methods and Resources entitled "A new branched proximity hybridization assay for the quantification of nanoscale protein-protein proximity" for publication in PLOS Biology. I have now obtained advice from the original reviewers and have discussed their comments with the Academic Editor. 

Based on the reviews, we will probably accept this manuscript for publication, assuming that you will modify the manuscript to address the remaining points raised by the reviewers. Please also make sure to address the data and other policy-related requests noted at the end of this email.

We expect to receive your revised manuscript within two weeks. Your revisions should address the specific points made by each reviewer. In addition to the remaining revisions and before we will be able to formally accept your manuscript and consider it "in press", we also need to ensure that your article conforms to our guidelines. A member of our team will be in touch shortly with a set of requests. As we can't proceed until these requirements are met, your swift response will help prevent delays to publication.

Sincerely,

Di Jiang, 

Associate Editor

PLOS Biology

ETHICS STATEMENT:

Please include an Ethics Statement subsection in the beginning of the Methods section. The Ethics Statements in the submission form and Methods section of your manuscript should match verbatim. Please ensure that any changes are made to both versions.

Please include the full name of the IACUC/ethics committee that reviewed and approved the animal care and use protocol/permit/project license in the Ethics Statement. Please also include an approval number.

Please include the specific national or international regulations/guidelines to which your animal care and use protocol adhered. Please note that institutional or accreditation organization guidelines (such as AAALAC) do not meet this requirement.

Reviewer remarks:

Reviewer #1: Zheng et al. have now resubmitted their paper after addressing comments by both reviewers. 

As previously, the authors claim on lines 19 and 71 a 400-fold amplification by the bPHA technique, now supported by a reference to information from the manufacturer, but the number represents a theoretical maximal signal amplification, and since no evidence is given that this amplification is actually achieved it would be prudent to describe this as a theoretical limit. Comparing panels 2B and 2D the amplified bPHA signals, where detection depends on binding by pairs of affinity reagents, seems to yield around ten-fold higher fluorescence than that of a single, unamplified Enh-Cy5 probe. Admittedly, the dependence for detection on binding by pairs of probes might reduce efficiency compared to assays that depends on binding by single reagents so this may not be a fair comparison for estimating signal amplification.

Throughout the manuscript the authors describe the RCA process as nonlinear and the bDNA techniques as linear, but it is unclear what they mean by this. Both RCA and bDNA would be expected to give rise to local amplification products in direct proportion to the proper starting molecules, in the case of RCA a DNA circle and for bPHA proximal target sequences for the two Z-DNA probes. In the new figure 2 GFP expression of the Ramos cells seems to vary over more than an order of magnitude in the bPHA experiment, but less than one order of magnitude for the PLA experiment, complicating estimation of correlation or linearity of responses.

In line 225 bPHA is said to correlate to GFP over three orders of magnitude differences of GFP expression, but this is not evident from figure 2 where in the bPHA experiment GFP varied over a little more than an order of magnitude. It would be prudent to point this out. 

This figure 2 is rendered a little confusing by the inclusion of zero values in the logarithmic axes of panels B, C, D, F and G. The zero values should be replaced by the proper numbers. This is also true for panels 1H and 3B,C and D, 4A and 5B, as well as for supplementary figures 2, 3, 4 and 6.

As a final point, were the TD05+ and – probes really used at 1 mM concentration as stated in line 339? This is a surprisingly high concentration.

Reviewer #2: All points raised have been adressed

---

## [Editor Report · Decision Letter 3]

14 Nov 2019

Dear Dr YANG,

On behalf of my colleagues and the Academic Editor, Ana J. Garcia-Saez, I am pleased to inform you that we will be delighted to publish your Methods and Resources in PLOS Biology. 

Early Version

PRESS 

Kind regards,

Sofia Vickers

Senior Publications Assistant

PLOS Biology

On behalf of, 

Di Jiang,

Associate Editor

PLOS Biology